# Bidirectional Interaction between Tetracyclines and Gut Microbiome

**DOI:** 10.3390/antibiotics12091438

**Published:** 2023-09-12

**Authors:** Jerzy Jaroszewski, Niles Mamun, Krzysztof Czaja

**Affiliations:** 1Department of Pharmacology and Toxicology, Faculty of Veterinary Medicine, University of Warmia and Mazury in Olsztyn, Oczapowskiego 13, 10-718 Olsztyn, Poland; jerzyj@uwm.edu.pl; 2Department of Biomedical Sciences, College of Veterinary Medicine, University of Georgia, Athens, GA 30602, USA; niles.mamun@uga.edu

**Keywords:** antibiotics, tetracyclines, gut microbiome, interaction

## Abstract

The escalating misuse of antibiotics, particularly broad-spectrum antibiotics, has emerged as a pivotal driver of drug resistance. Among these agents, tetracyclines are widely prescribed for bacterial infections, but their indiscriminate use can profoundly alter the gut microbiome, potentially compromising both their effectiveness and safety. This review delves into the intricate and dynamic interplay between tetracyclines and the gut microbiome, shedding light on their reciprocal influence. By exploring the effects of tetracyclines on the gut microbiome and the impact of gut microbiota on tetracycline therapy, we seek to gain deeper insights into this complex relationship, ultimately guiding strategies for preserving antibiotic efficacy and mitigating resistance development.

## 1. Introduction

Defined daily doses (DDD) is a metric used to compare the usage of drugs based on what constitutes a dose for each particular drug. In 2000, the world consumed 21.1 billion DDDs of antibiotics [1]. This figure increased by an alarming 90.5% to 40.2 billion DDDs in 2018 [2]. Rates of antibiotic consumption are high in high-income and upper-middle-income countries in North America, Europe, and the Middle East, while very low in less affluent regions of sub-Saharan Africa and Southeast Asia. This gap is beginning to close, however, as antibiotic consumption rates have been relatively stable in high-income countries while there was a 76% increase in consumption rates from 2000–2018 in lower-income and middle-income countries [2]. When considering the increase in antibiotic usage, this growing problem appears to most directly relate to excessive prescription. The Centers for Disease Control and Prevention (CDC) estimates that at least 30% of outpatient antibiotic prescriptions are completely unnecessary [3]. With roughly one in three antibiotic prescriptions being unwarranted, the answer to this problem would seem evident: prescribe fewer antibiotics; however, for many reasons, it is not that simple. Depending on the physician’s tolerance of uncertainty, the inability to make a clear diagnosis prompts the use of antibiotics to avert the risk of not prescribing antibiotics in instances where this should have occurred [4]. Another critical factor driving inappropriate antibiotic prescriptions is patient expectation, as many patients have become accustomed to treating any infection with antibiotics, and physicians may succumb to these pressures by fulfilling their patients’ requests [5].

Beyond the amount of antibiotics being prescribed and consumed, the kind of antibiotic becomes an additional important factor. There are two general types of antibiotics: narrow-spectrum and broad-spectrum. Narrow-spectrum antibiotics target a specific group of bacteria, while broad-spectrum antibiotics work against a wider variety of bacteria [6]. In 2017, the World Health Organization (WHO) introduced the ACCESS, WATCH, and RESERVE classifications of antibiotics in an effort to progress antibiotic stewardship. The ACCESS group of antibiotics should be the first treatment choice and consists of narrow-spectrum agents. The WATCH group of antibiotics are broad-spectrum agents with a higher propensity to induce antibiotic resistance. The RESERVE group is made up of last-resort antibiotics to prevent resistance to and maintain the effectiveness of these specific antibiotics [7,8]. The global per-capita consumption of ACCESS group antibiotics increased by 26.2% between 2000 and 2015. While concerning on its own, the increase in ACCESS usage pales in comparison to the increase in WATCH antibiotics usage. From 2000 to 2015, WATCH antibiotic global per-capita consumption nearly doubled, increasing by 90.9% [9]. Proponents of antibiotic stewardship advocate for most antibiotic prescriptions to fall under the ACCESS category (narrow-spectrum), but at the current rate, WATCH antibiotics (broad-spectrum) will soon make up most of all prescriptions.

The rapid growth in broad-spectrum antibiotic usage again does not stem from necessity but rather misuse. A study monitoring the prescriptions for outpatient parenteral antibiotic therapy found that up to 50% of broad-spectrum prescriptions could have been narrowed [10]. However, even in instances where narrow-spectrum antibiotics are more justified, physicians still have the tendency to prescribe broad-spectrum antibiotics [11]. This could be because prescribing practices are often habitual, and decisions may be made using treatments most familiar to the prescriber, prompting the use of all-encompassing broad-spectrum antibiotics [12].

Tetracyclines are known to have a broad spectrum of activity against many different bacterial species, including both Gram-positive and Gram-negative bacteria. They possess antimicrobial activity by binding to the 30S ribosomal subunit, which prevents the binding of aminoacyl-tRNA to the ribosome and consequently inhibits bacterial protein synthesis in growing or multiplying pathogenic organisms [13,14,15]. However, tetracyclines can also affect the gut microbiome, which can have negative consequences for the patient being treated.

The human microbiome is made up of the microbes that live within and on the human body. Biological interaction with the immune system and these organisms over time have allowed for the formation of a symbiotic relationship, benefitting the human host and the colonizers [16]. Humans have many microbiomes, such as the skin, the mucosa, the gastrointestinal tract, the mammary gland, and the urogenital tract [17]. The gut microbiome is a complex ecosystem consisting of trillions of microorganisms that play a vital role in human health. The gut microbiota is involved in many physiological processes, including digestion, nutrient absorption, and metabolism [18]. Tetracyclines, as broad-spectrum antibiotics, can alter the composition and diversity of the gut microbiome by selectively targeting certain bacterial species while leaving others relatively unaffected. This can lead to the overgrowth of opportunistic pathogens, which can cause severe dysbiosis [19].

The gut microbiome can also impact tetracycline therapy [20]. Gut microorganisms can affect drug metabolism and absorption, potentially altering the efficacy and side effects of tetracyclines. For example, some gut bacteria can produce enzymes that modify tetracyclines, making them less effective against certain bacterial species [21]. Gut microbiota can also play a role in tetracycline-induced toxicity. For example, tetracyclines can cause liver toxicity, and the gut microbiome can modulate liver metabolism, potentially exacerbating this side effect.

We aimed to review current research on bidirectional interaction between tetracyclines and the gut microbiome. Tetracyclines can have negative effects on the gut microbiome, and the gut microbiota can impact the efficacy and side effects of tetracycline therapy. Understanding this bidirectional relationship between tetracyclines and the intestinal microbiome is essential for optimizing antibiotic therapy, preserving gut health, and advancing our knowledge of the broader implications for human health and antibiotic resistance.

## 2. General Information about Tetracyclines Used in Humans

Tetracyclines are a large group of natural and semi-synthetic antibiotics with bacteriostatic activity, but in high concentrations, they can also have bactericidal activity [22]. Drugs from this group were discovered and first used in medicine in the middle of the last century [23]. The first natural tetracycline discovered was chlortetracycline (aureomycin), isolated in 1948 from a culture of *Streptomycces aureofaciens* [24]. Soon after this discovery in 1949, oxytetracycline (terramycin) was isolated from *Streptomyces rimosus* [25]. Other natural drugs from this group were tetracycline produced by *Streptomycces* spp. (in 1953) and demeclocycline isolated from *Streptomycces aureofaciens mutant* [26]. Although tetracycline was discovered later than chlortetracycline and oxytetracycline, it is still considered the parent compound for nomenclature purposes [27]. As a result of chemical modifications of natural tetracyclines, semi-synthetic tetracyclines were synthesized, constituting the second generation of these antibiotics. These include rolitetracycline, lymecycline, clomocycline, methacycline, doxycycline, and minocycline [28,29]. A special biotechnological achievement in the development of tetracyclines was the introduction of structural changes in the minocycline molecule (addition of 9-tert-butylglycyl amide at carbon 9 in the D ring), this being the method by which tigecycline was obtained, the first therapeutically used tetracycline (glycylcycline) of the third generation [15,30]. In 2018, the Food and Drug Administration (FDA) approved three new tetracyclines for human use: sarecycline, omadacycline, and eravacycline [31]. This is the only case of the group of “old antibiotics”, for which 70 years from the use of the first drugs, a few new active substances have been introduced.

Tetracycline molecules comprise a linear fused tetracyclic nucleus (rings designated A, B, C and D) to which a variety of functional groups are attached. While all tetracyclines have a common structure, they differ from each other in the presence of the chloride, methyl, and hydroxyl groups. These modifications do not change their broad antibacterial activity but do affect pharmacokinetic properties such as half-life and binding to proteins [15,29].

In general, tetracyclines can be divided into three groups based on their pharmacokinetic and antibacterial properties. Group 1 consists of the older agents (tetracycline, oxytetracycline, chlortetracycline, demeclocycline, lymecycline, and rolitetracycline), which have reduced absorption and are less lipophilic than newer drugs from Group 2; all can be administered orally except rolitetracycline. Group 2 consists of doxycycline and minocycline, which are better absorbed from the gastrointestinal tract (GI) and are 3–5 times more lipophilic than drugs from Group 1. Group 3 consists of tigecycline, sarecycline, omadacycline, and eravacycline, which are active against many bacteria with acquired resistance to older tetracyclines [29,32].

The most important information on the pharmacokinetics (bioavailability, main site of metabolism, residence time in the body, and route of excretion) of tetracyclines used in humans is presented in Table 1. The data presented in the table indicate that tetracyclines are absorbed to varying degrees from the GI, are metabolized to a small extent in the liver (apart from minocycline), and are excreted with urine and feces regardless of the route of administration.

Bioavailability refers to the fraction of an administered drug that reaches the systemic circulation in an active form. The bioavailability of tetracyclines from the GI can be influenced by several factors, including the drug’s chemical structure, route of administration, the presence of food in the stomach, potential food–drug interactions, and the composition of the gut microbiome [39,40,41]. Tetracyclines are primarily absorbed in the small intestine and, to maximize their bioavailability, are typically administered on an empty stomach at least one hour before or two hours after meals. It is well documented that the bioavailability of orally administered tetracyclines is reduced in the presence of food, dairy products, and agents containing divalent and trivalent cations, which necessitates the spacing of drug administration. However, literature data indicate that there are significant differences in this respect depending on the group of tetracyclines. All tetracyclines belonging to Group 1 form insoluble complexes with calcium, magnesium, iron, and aluminum, which markedly reduces absorption [42]. In contrast, doxycycline–metal ion complexes are unstable at acid pH; therefore, more doxycycline than minocycline enters the duodenum for absorption [32].

Moreover, food has less of an effect on the absorption of doxycycline and minocycline than on the absorption of drugs from Group 1 [32]. Tigecycline and eravacycline are administered intravenously; therefore, potential interactions at the stage of absorption from the gastrointestinal tract should not be significant for the therapeutic effect. However, it should be borne in mind that due to biliary excretion, they may affect the intestinal microbiome. In the case of sarecycline, the results of the current pharmacokinetics studies demonstrate that sarecycline can be administered using a weight-based dosing regimen and can be taken with or without food without a clinically relevant impact on efficacy [43]. In contrast, consuming food, especially high-fat meals and dairy products, during dosing significantly reduces omadacycline bioavailability [41,44].

The main mechanism of action of tetracyclines is related to the inhibition of bacterial protein synthesis by preventing the association of aminoacyl-tRNA with the bacterial ribosome [15]. They also have anti-inflammatory effects based on multiple reported biologic properties, including inhibition of matrix metalloproteinases, suppression of IL-8, TNFα, and IL-6 gene expression from neutrophils and macrophages, suppression of hydrolases, and scavenging of reactive oxygen species [45,46]. However, such mechanisms of action, together with the broad-spectrum activity, may disrupt the intestinal microflora, leading to dysbiosis, which contributes to the increased incidence of gastrointestinal side effects, diarrhea, and intestinal fungal overgrowth, particularly in patients receiving prolonged oral antibiotic therapy [47,48,49].

Due to the broad-spectrum activity against many pathogens, including Gram-positive and Gram-negative bacteria, spirochetes, chlamydia, leptospira, mycoplasma and rickettsia tetracyclines (tetracycline, doxycycline, minocycline, tigecycline, sarecycline, omadacycline, eravacycline and demeclocycline) are widely used in medical practice (tetracycline, doxycycline, minocycline, tigecycline, sarecycline, omadacycline, eravacycline and demeclocycline). Tetracycline and demeclocycline are used to treat pneumonia and other respiratory tract infections and certain infections of the skin, eye, lymphatic, intestinal, genital and urinary systems. Doxycycline indications include upper respiratory, skin, or soft tissue infections, gonorrhea and syphilis in penicillin-allergic patients, non-gonococcal urethritis, acute pelvic inflammatory disease, epididymitis, oorchitis, Lyme disease, and as prophylaxis against traveler’s diarrhea. Current indications of minocycline are for therapy of acne, gonorrhea, syphilis, non-gonococcal urethritis, chlamydial infections, cholera, leprosy, and the meningococcal carrier state. Tigecycline indications include treatment of community-acquired pneumonia and complicated skin, tissue and intra-abdominal infections due to sensitive organisms [50]. Sarecycline is approved for the treatment of acne vulgaris [34]. Omadacycline is used to treat adults with acute bacterial skin and skin structure infections (ABSSSI) and community-acquired bacterial pneumonia (CABP) [36]. Eravacycline has been indicated for the treatment of complicated intra-abdominal infections in adults [51].

The research discussed in this chapter clearly indicated that tetracyclines, regardless of the drug used and route of administration, are present in the digestive tract and can affect its intestinal flora. However, at present, there are no comparative data clearly indicating and describing the differences in the effect of individual drugs on the intestinal microbiome.

## 3. Tetracycline Effects on the Gut Microbiome

Tetracyclines can have unintended consequences on the gut microbiome, which is the collection of microorganisms that live in the GI [52]. The gut microbiome is a complex ecosystem of microorganisms that inhabit the gastrointestinal tract. It includes bacteria, fungi, viruses, and other microorganisms that play a crucial role in human health. The gut microbiome is involved in the digestion and absorption of nutrients, the production of vitamins and other essential compounds, and the maintenance of immune function [39,40,41]. Alterations in the composition of gut microbiota can be induced by several exogenous factors, with antibiotic abuse being the most powerful [53,54]. Other factors, including stress, radiation, gastrointestinal infections, and dietary changes, can induce dysbiosis [55]. There are three kinds of dysbiosis: loss of beneficial bacteria, overgrowth of harmful bacteria, and loss of diversity in the gut flora, and in most cases of dysbiosis, all three events occur simultaneously [56]. This chapter will next discuss how tetracyclines can affect the gut microbiome and the potential implications for human health.

According to the WHO AWaRe classification, tetracyclines have been assigned to ACCESS (tetracycline and doxycycline) WATCH (chlortetracycline, demeclocycline, lymecycline, metacyline, minocycline for oral use, oxytetracycline, penimepicyline, rolitetracycline, and sarecycline) and RESERVE (eravacycline, minocycline for intravenous use, omadacycline and tigecycline [8]. The influence of ACCESS antibiotics on gastrointestinal dysbiosis is presented in Table 2. In a study treating ten healthy participants with doxycycline, decreases in *Enterobacteriaceae*, *Enterococcus* spp., *E. coli*, and *Streptococcus* spp. were seen, while *Fusobacterium* spp. was eliminated. Bacteria levels returned to pre-antibiotic abundance nine days after treatment [57,58]. Another study comparing the effects of doxycycline plus probiotic treatment compared to just probiotic treatment found that a decrease in *Bifidobacterium* diversity occurred in the antibiotic-treated group as well as an increase in tetracycline-resistant *Bifidobacterium* [58,59]. Despite being an ACCESS antibiotic, doxycycline shows the propensity to induce gut dysbiosis, but patients normally recover within a month of treatment [58]. In a study comparing the conventional gut community to the tetracycline-treated gut community in worker honeybees, the antibiotic-treated workers possessed a lower absolute abundance of gut bacteria. While *Lactobacillus*, *Frischella*, *Commensalibacter*, *Bartonella*, *Snodgrassella*, and *Gilliamella* were the dominant bacteria in both populations, the tetracycline-treated bees saw a decrease in these bacteria [60]. Another study focusing on tetracycline’s effect on honeybees monitored alterations to the gut microbiome and survivorship post-treatment. This study found that *Bifidobacterium*, Firm-4, Firm-5, *Snodgrassella alvi*, Alpha 2.1, *Frischella perrara*, *Lactobacillus kunkeei*, and *Bartonella apis* all decreased at some point after treatment. *Serratia*, *Holomonadaceae*, and *Gilliamela apicola* were elevated after treatment. Furthermore, it was found that tetracycline-treated bees had a decreased survivorship [61].

Oxytetracycline and orally administered minocycline are listed by the WHO as WATCH antibiotics. The influence of WATCH antibiotics on gastrointestinal dysbiosis is presented in Table 3. In a study that administered 4-epi-oxytetracycline, one of the main oxytetracycline metabolites, on rats to view changes in their gut microbiome and resistome, significant changes in gut composition were seen. In high-dose male rats, *Bifidobacteriaceae*, *Enterococcaceae*, and *Actinomycetacaea* significantly increased, while *Lactobacillaceae*, *Aerococcaceae*, *Helicobacteraceae*, and *Pasteurellaceae* decreased in abundance. Similar results were seen in female rats, except *Bifidobacteriaceae* levels did not significantly increase. Resistance also saw a significant increase following treatment. The antibiotic resistance genes (ARGs) *tetO* and *tetQ* were more abundant in high-dose treated rats compared to control rats. Composition started recovering after treatment but still displayed abnormalities two weeks after treatment. 4-epi-oxytetracycline was found in blood and tissue samples two weeks after treatment, and metabolism significantly changed in treatment groups [63]. In a study determining how a single treatment of oxytetracycline affected the gut microbiome of Nile Tilapia, oxytetracycline-treated fish saw decreases in *Actinobacteria*, *Lamia*, *Aeromonas*, *Pseudomonas*, *Reyranella*, *Nocardioides*, *Mycobacterium*, *Smaragdicoccus*, *Pedomicrobium*, Chlamydiae, Verrucomicrobia, *Gematta*, *and Planctopirus.* Increases were seen in *Plesiomonas*, *Aquicella*, *Hyphomicrobium*, *Actinobacteria*, *Bacteroidetes*, *Chlroflexi*, *Firmicutes*, *Acidobacteria*, *Cetobacterium*, and *Macellibacteroides* [64]. A study examining how oxytetracycline impacts the gut microbiome of zebrafish found that *Cetobacterium*, *Aeromonas*, *Shewanella*, *Plesiomonas*, and *Enterobacterales* all decreased in antibiotic-treated fish. On the other hand, *Mesorhizobium*, *Rhizobiaceae*, *Pseudomonas*, *Variovorax*, *Shewanella*, and *Bacteroides* all increased after treatment [65].

In a study examining how acne treatment using oral minocycline impacted the gut and skin microbiota, researchers found significant changes when comparing the gut bacteria after treatment to before. No antibiotic bacteria were enriched after antibiotic treatment; however, patients were depleted in *Lactobacillus salibarius*, *Bifidobacterium adolescentis*, *Bifidobacterium pseudolongum*, and *Bifidobacterium breve* [49]. Two of these bacteria, *B. adolescentis* and *L. salivarius*, are lactic acid bacteria proven to have probiotic effects [66,67]. Another study found that minocycline caused a 6.1-fold reduction in *Lactobacillus* spp., a five-fold increase in *Enterobacteriaceae*, and a 4.1-fold increase in *Enterococcus* spp. Bacteria returned to pre-antibiotic levels during the recovery phase of the experiment except for *Enterococcus* spp., which remained high [68]. Accumulation and persistence of *Enterococcus* spp. following minocycline treatment is concerning as *tet(M*) is the most common tetracycline-resistant determinant among enterococci, and the Tet(M) protein has the potential to reduce minocycline susceptibility [69,70]. One study that viewed how minocycline altered the rat microbiome saw an increase in the butyrate-producing *Clostridiales* family XIII and *Lachnospiraceae* families. The study observed phenotypic and behavioral changes relating to anxiety after minocycline treatment in certain populations [71]. Another study observing changes in the rat microbiome after minocycline treatment also saw increases in *Lachnospiraceae* as well as *Porphyromonadaceae* after treatment. *Lactobacillus* and *Blautia* were observed to decrease after treatment. The researchers found that minocycline treatment prevented and reversed diet-induced impairment of spatial recognition memory in rats [72].

**Table 3 antibiotics-12-01438-t003:** Effect of WATCH antibiotics on gastrointestinal dysbiosis.

Antibiotic	Experimental Model	Bacteria Decreased	Bacteria Increased	Time for Gut to Recover	Long Term Impacts
Oxytetracycline [63]	Wistar rat	*Lactobacillaceae*, *Aerococcaceae*, *Helicobacteraceae*, and *Pasteurellaceae*	*Bifidobacteriaceae*, *Enterococcaceae*, and *Actinomycetacaea*	Composition started recovering after treatment but still displayed abnormalities two weeks after treatment	4-epi-oxytetracycline was found in blood and tissue samples two weeks after treatment, ARGs increased, and metabolism significantly changed in treatment groups
Oxytetracycline [64]	Nile Tilapia	*Actinobacteria*, *Lamia*, *Aeromonas*, *Pseudomonas*, *Reyranella*, *Nocardioides*, *Mycobacterium*, *Smaragdicoccus*, *Pedomicrobium*, Chlamydiae, Verrucomicrobia, *Gematta*, *Planctopirus*	*Plesiomonas*, *Aquicella*, *Hyphomicrobium*, *Actinobacteria*, *Bacteroidetes*, *Chlroflexi*, *Firmicutes*, *Acidobacteria*, *Cetobacterium*, *Macellibacteroides*	Not Applicable	Disruption of microbiome could act as a pressure in resistance development in the recovered community
Oxytetracycline [65]	Zebrafish	*Cetobacterium*, *Aeromonas*, *Shewanella*, *Plesiomonas*, *Enterobacterales*	*Mesorhizobium*, *Rhodobacteraceae*, *Rhizobiaceae*, *Pseudomonas*, *Variovorax*, *Shewanella*, *Bacteroides*,	Up to 1 month after treatment	Post-exposure changes in gut flora were observed
Minocycline (oral) [49]	Human	*Lactobacillus salivarius*, *Bifidobacterium adolescentis*, *Bifidobacterium pseudolongum*, and *Bifidobacterium breve*	*Bacteroidetes*	Not Applicable	Not Applicable
Minocycline (oral) [68]	Human	*Lactobacillus* spp.	*Enterobacteriaceae* and *Enterococcus* spp.	Almost entirely recovered 3 weeks after treatment	*Enterococcus* spp. remained high Several families failed to recover
Minocycline (oral) [71]	Rat	Not Applicable	*Lachnospiraceae*, *Clostridiales* Family XIII	Not Applicable	Antidepressant effects observed depending on traits and sex
Minocycline (oral) [72]	Rat	*Lactobacillus*, *Blautia*	*Lachnospiraceae*, *Porphyromonadaceae*	Not Applicable	Prevented and reversed impairments in spatial recognition memory caused by diet

The RESERVE group of antibiotics such as omadacycline should be used sparingly to avoid resistance and maintain antibiotic efficacy. In a study investigating how omadacycline impacts the human gut microbiome, bifidobacterial, lactobacilli, *Bacteroides fragilis*, and enterococci populations were seen to decrease (Table 4). Lactose-fermenting *Enterobacteriaceae* populations increased [73]. Beyond disrupting the microbial balance, growth in lactose-fermenting *Enterobacteriaceae* in the gastrointestinal tract increases flatulence as a result of elevated levels of gas production [74]. *Enterobacteriaceae* are also known to be pro-inflammatory, meaning the overgrowth of these populations could result in gut inflammation [75].

Tetracyclines, like all antibiotics, can affect the beneficial bacteria that make up the gut microbiome [39,76]. This can lead to a decrease in gut microbiome diversity, which has been linked to a range of health problems, including inflammatory bowel disease, obesity, and type 2 diabetes. One way that tetracyclines decrease gut microbiome diversity is by selectively inhibiting the growth of certain bacterial species [13,14,77]. However, they are not equally effective against all bacterial species. Some species, such as those that are Gram-negative or anaerobic, are less susceptible to tetracyclines, while others, such as those that are Gram-positive or aerobic, are more susceptible. Another way that tetracyclines decrease gut microbiome diversity is by disrupting the balance of the microbial community. The gut microbiome is composed of many different species of bacteria that interact with each other in complex ways. When antibiotics are administered, they can disrupt these interactions, leading to changes in the microbial community structure [78,79,80]. This can result in a decrease in diversity as certain species become dominant and others are eliminated or suppressed. Furthermore, tetracyclines can also promote the growth of antibiotic-resistant bacteria. This can lead to the proliferation of certain species that are resistant to tetracyclines, further reducing the overall diversity of the gut microbiome [78,81,82].

Studies have shown that tetracyclines can not only have short-term effects on the gut microbiome but also long-term effects. Short-term effects include changes in the composition and function of the gut microbiome during tetracycline treatment, while long-term effects can persist even after the treatment has ended. In the short term, within hours of administration, tetracyclines can significantly reduce the abundance of many bacterial species, including those that are essential for gut health, such as *Lactobacillus* and *Bifidobacterium*. This reduction in diversity can have negative consequences for gut function, such as impaired nutrient absorption, inflammation, and an increased risk of colonization by opportunistic pathogens [83,84]. The study showed that ciprofloxacin (a drug from the group of fluoroquinolones with a broad spectrum of activity against aerobic bacteria) rapidly decreased microbial diversity and altered the composition of the gut microbiome within hours of administration. Furthermore, the effects of ciprofloxacin persisted for several weeks after the antibiotic was discontinued, suggesting long-term impacts on the gut microbiome. In the long term, the effects of antibacterial drugs, including tetracyclines, on the gut microbiome can be more profound than previously realized [79]. Even after the antibiotics have been discontinued, some studies have shown that the gut microbiome may not fully recover for months or even years. This may be due in part to the development of antibiotic resistance among some gut bacteria, which can persist even after antibiotic exposure has ceased. 

Furthermore, repeated antibiotic exposure over time can lead to a cumulative decrease in microbial diversity, making the gut microbiome more vulnerable to perturbations and potentially increasing the risk of chronic health conditions, such as inflammatory bowel disease and metabolic disorders. The long-term effects of tetracyclines on the gut microbiome may also depend on the age of the patient and other factors, such as diet, genetics, and environmental exposures [85,86,87]. For example, early-life exposure to antibiotics has been shown to have lasting effects on the gut microbiome and may increase the risk of developing certain diseases later in life. Similarly, certain dietary patterns, such as a high-fat, low-fiber diet, may exacerbate the negative effects of antibiotics on the gut microbiome. In the long term, repeated antibiotic exposure can lead to a cumulative decrease in diversity and potentially increase the risk of chronic health conditions. Therefore, it is important to carefully consider the risks and benefits of antibiotic therapy and to explore alternative strategies, such as probiotics and dietary interventions, to support the gut microbiome during and after antibiotic treatment.

During tetracycline treatment, there is a decrease in the abundance of susceptible bacteria, which can lead to an overgrowth of antibiotic-resistant bacteria [88,89,90,91]. This overgrowth can result in the development of antibiotic-associated diarrhea, which is a common side effect of tetracycline treatment. Additionally, tetracyclines can also affect the metabolism of bile acids, which can impact the absorption of nutrients and the efficacy of certain drugs that are metabolized in the liver. When susceptible bacteria are eliminated during tetracycline treatment, the selective pressure on the gut microbiome can favor the growth of resistant bacteria, which may have a survival advantage in the absence of competition from susceptible strains. The overgrowth of antibiotic-resistant bacteria during tetracycline treatment can have serious implications for human health. These bacteria can cause infections that are difficult to treat with standard antibiotics, leading to prolonged illness, hospitalization, and even death. Moreover, the spread of antibiotic-resistant bacteria can contribute to the global health threat of antibiotic resistance, making it more challenging to treat a range of bacterial infections.

Tetracyclines, in some cases, can selectively target pathogenic bacteria while leaving beneficial bacteria relatively unaffected. For example, tetracyclines have been shown to have a preferential effect on Gram-negative bacteria, such as *Salmonella* and *Escherichia coli*, while having little effect on Gram-positive bacteria such as *Lactobacillus* and *Bifidobacterium* [15]. This selective targeting can result in the reduction of harmful bacteria in the gut, which can be beneficial for gut health. Tetracyclines have also been shown to have immunomodulatory effects in the gut. Specifically, tetracyclines have been shown to reduce the production of pro-inflammatory cytokines and increase the production of anti-inflammatory cytokines in the gut [92,93]. Moreover, the results show that tetracyclines were effective in counteracting most of the markers found altered in DNBS-colitis and increased mucosal protection through the upregulated expression of CCL2, miR-142, and miR-375, leading to improved microbial-derived signaling and mucosal protection [84]. These findings suggest the potential of immunomodulatory tetracyclines to prevent inflammation-associated tissue damage in acute intestinal inflammation.

The research discussed in this chapter provides clear evidence that the use of tetracyclines can have unintended consequences on the gut microbiome, which can lead to a decrease in microbial diversity, disruption of microbial community structure, and promotion of antibiotic-resistant bacteria. Mechanisms of dysbiosis caused by ACCES, WATCH and RESERVE tetracyclines are presented in Table 5, Table 6 and Table 7, respectively. While tetracyclines are known to have a negative impact on the gut microbiome, they can also have beneficial effects under certain circumstances. However, it is important to note that the use of antibiotics should always be judicious, and their potential benefits and risks should be carefully weighed. Thus, understanding the interactions between tetracyclines and the gut microbiome is crucial for optimizing their therapeutic efficacy and minimizing the risk of adverse effects.

## 4. How Can the Gut Microbiome Alter Tetracycline Treatment?

The gut microbiome plays an important role in the metabolism of drugs, including antibiotics such as tetracyclines. Tetracyclines are metabolized by the liver and excreted in the bile, where they can be reabsorbed in the small intestine. Recent studies have shown that gut bacteria can metabolize tetracyclines into biologically active compounds, impacting the pharmacokinetics and efficacy of these antibiotics [94,95]. For example, an increase in the abundance of certain gut bacteria, such as *Enterococcus faecalis*, can reduce the efficacy of tetracycline [96]. In particular, certain bacterial species can produce enzymes that modify antibiotics, making them less available [21]. For example, some gut bacteria can produce enzymes that inactivate tetracyclines by oxidizing them, which can lead to decreased bioavailability and reduced distribution of the antibiotic throughout the body. The gut bacterium *Bacteroides fragilis* can produce an enzyme called tetracycline destructase, which can break down tetracyclines into inactive metabolites [21]. Similarly, other bacterial species such as *Escherichia coli*, *Enterococcus faecalis*, and *Pseudomonas* can modify tetracyclines and reduce their bioavailability. There are several bacterial species that can produce enzymes that modify tetracyclines. Tet(X)-producing bacteria carry the *tet(X)* gene, which encodes a flavin-dependent monooxygenase enzyme that can inactivate tetracyclines. Examples of bacteria that produce Tet(X) include *Bacteroides fragilis*, *Escherichia coli*, and *Acinetobacter baumannii* [97,98,99,100,101]. Tet(M)-producing bacteria carry the *tet(M)* gene, which encodes a ribosomal protection protein that can protect the bacterial ribosome from the inhibitory effects of tetracyclines. Examples of bacteria that produce Tet(M) include *Streptococcus pneumoniae*, *Staphylococcus aureus*, and *Enterococcus faecalis* [102,103]. Tet(B)-producing bacteria carry the *tet(B)* gene, which encodes a membrane-associated transporter protein that can pump tetracyclines out of the bacterial cell. Examples of bacteria that produce Tet(B) include *Pseudomonas aeruginosa*, *Klebsiella pneumoniae*, and *Enterobacter cloacae* [104,105,106,107]. 

Moreover, changes in the composition of the gut microbiome can impact tetracycline distribution by altering the permeability of the intestinal barrier [108,109,110]. The intestinal barrier acts as a protective layer that prevents the passage of harmful substances into the bloodstream. However, changes in the gut microbiome can cause alterations to the structure and function of the intestinal barrier, resulting in increased permeability. This can lead to increased absorption of tetracyclines and enhanced distribution throughout the body. Tetracyclines are primarily excreted through the bile and can be reabsorbed in the intestine as a result of enterohepatic recirculation [111]. The gut microbiome can influence the composition of the bile, altering the concentration of tetracyclines and other compounds excreted in the bile [112,113,114]. Moreover, it has been suggested that alteration of the gut microbiome led to changes in bile acid composition, which in turn affected the expression of liver enzymes involved in tetracycline metabolism [112]. Therefore, microbiome-induced changes in bile composition can affect the overall bioavailability of tetracyclines and their distribution in the body. Dysbiosis can also affect the absorption and distribution of antimicrobial agents in the body [115]. It has been shown that gut microbiota dysbiosis promotes bile acid homeostasis disbalance and disturbed gut barrier by increasing the expression of tight junction proteins and hepatic inflammatory cytokine secretion in the liver [116].

It is also suggested that the gut microbiome can impact the metabolism of tetracyclines in the liver. Tetracyclines are metabolized in the liver through the cytochrome P450 enzyme system. However, the gut microbiome can produce metabolites that can compete with tetracyclines for metabolism by the cytochrome P450 enzymes [117,118,119,120,121], resulting in decreased clearance of the antibiotic and increased distribution throughout the body. The gut microbiome can also influence tetracyclines metabolism through alterations in the expression of drug-metabolizing enzymes in the liver [122,123]. For example, studies have shown that the gut microbiome can modulate the expression of cytochrome P450 enzymes, which play a key role in drug metabolism in the liver. Changes in the expression of these enzymes can lead to alterations in the pharmacokinetics of tetracyclines and other drugs.

Measures can be taken to mitigate the effects tetracyclines and the gut microbiome have on each other. Introducing tetracyclines to the gut selects for tetracycline resistance genes, but supplementation of probiotics reduces the number of antibiotic resistance genes in the gut microbiome [124]. Probiotics are live microorganisms that can restore the balance of the gut microbiome, further mitigating the effects of tetracyclines; however, the effectiveness of probiotics in preventing or treating tetracycline-induced dysbiosis is still under investigation [125]. High-fiber diets prevent loss in microbiome diversity and reduce antibiotic-induced dysbiosis symptoms, while high-glucose diets exacerbate dysbiosis [126]. This indicates that consuming a diet high in fiber and limiting glucose consumption reduces the magnitude of effects tetracyclines would have on the microbiome. Limiting fat consumption should help to maintain tetracycline efficacy, as high-fat diets have been shown to reduce antibiotic efficacy [127].

## 5. Conclusions

The research discussed in this review shows that the interaction between tetracyclines and the gut microbiome is bidirectional, with the drugs affecting the composition and function of the microbiome and the microbiome influencing how the drugs are metabolized and their efficacy (Figure 1). On the one hand, tetracyclines can alter the gut microbiome diversity, leading to the overgrowth of opportunistic pathogens, affect the abundance and composition of specific bacterial taxa, and alter the metabolic activity of the microbiome by affecting the expression of genes involved in various metabolic pathways. This can have downstream effects on the production of metabolites that influence host physiology. On the other hand, the gut microbiome can influence how tetracyclines are metabolized and their efficacy. Conversely, some bacteria in the gut may enhance the activity of tetracyclines by producing enzymes that convert inactive forms of the drug into active ones. Additionally, the gut microbiome can influence the absorption and distribution of tetracyclines in the body by modulating the expression of drug transporters and metabolic enzymes in the gut epithelium and liver. The clinical implications of this complex interplay between tetracyclines and the gut microbiome are profound, emphasizing the necessity for a more nuanced understanding of the gut microbiome’s role in guiding antibiotic therapy. Recognizing the reciprocal dynamics at play could help optimize treatment outcomes, minimize resistance development, and preserve the efficacy of tetracyclines and other antibiotics. Further research is needed to fully elucidate the complex interplay between tetracyclines and the gut microbiome, including the mechanisms by which tetracyclines affect bacterial populations and how these changes can impact drug efficacy and toxicity. By better understanding the interaction between tetracyclines and the gut microbiome, we can identify strategies for improving their clinical use and combatting antibiotic resistance.

## Figures and Tables

**Figure 1 antibiotics-12-01438-f001:**
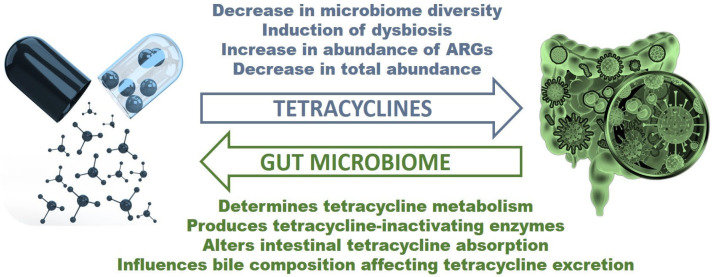
Schematic overview of bidirectional interaction between tetracyclines and the gut microbiome. (ARGs—antibiotic resistance genes).

**Table 1 antibiotics-12-01438-t001:** Pharmacokinetics of tetracyclines used in humans.

Active Substance [Reference]	Administration Routes	Bioavailability from GI (%)	Metabolism	Biological Half-Fife (h)	Excretion
Tetracycline [29,32]	Oral, topical	75–88	Minimally metabolized	6–11	Renal, feces
Oxytetracycline [29,32]	Oral, ophthalmic	58	Not metabolized	6–9.2	Renal, feces
Chlortetracycline [29,32]	Oral, topical	25–30	Not metabolized	5.6–9	Renal, biliary
Demeclocycline [29,32]	Oral	60–80	Hepatic	10–17	Renal
Lymecycline [29]	Oral	100	-	10	Renal
Rolitetracycline [32]	Intravenous	-	Not metabolized	5.8	Renal
Doxycycline [29,32]	Oral, intravenous	80–100	Not metabolized	15–25	Feces, renal
Minocycline [29,32]	Oral	100	Hepatic	11–18	Renal, feces
Tigecycline [29,33]	Intravenous	- *	Not metabolized	42.4	Biliary, renal
Sarecycline [34]	Oral	-	Minimally metabolized	21–22	Rena, feces
Omadacycline [35,36]	Oral, intravenous	34.5	Not metabolized	16.8	Feces, renal
Eravacycline [37]	Intravenous	28	Minimally metabolized	48	Biliary, renal

* The bioavailability of tigecycline after per os administration in turkey was 0.97 ± 0.57% [38]; GI—gastrointestinal tract.

**Table 2 antibiotics-12-01438-t002:** Effect of ACCESS antibiotics on gastrointestinal dysbiosis.

Antibiotic	Experimental Model	Bacteria Decreased	Bacteria Increased	Time for Gut to Recover	Long Term Impacts
Doxycycline [57]	Human	*Enterobacteriaceae*, *Enterococcus* spp., *Escherichia coli*, *Streptococcus* spp., and *Fusobacterium* spp.	Not Applicable	9 days after treatment	Not Applicable
Doxycycline [59]	Human	*Bifidobacterium*	Not Applicable	Not Applicable	Increase in tetracycline resistance
Doxycycline [62]	Human	*Escherichia coli*, *Enterococcaceae*	Not Applicable	4 weeks after 16-week treatment	Increased doxycycline resistance
Tetracycline [60]	Honeybee	*Lactobacillus*, *Frischella*, *Commensalibacter*, *Bartonella*, *Gilliamella*, *Snodgrassella*	Not Applicable	Did not recover	Gut microbiota did not recover in treated bees. This could harm the colony as contact with hive mates is a major contributor to bee microbiota
Tetracycline [61]	Honeybee	*Bifidobacterium*, Firm-4, Firm-5, *Snodgrassella alvi*, Alpha 2.1, *Frischella perrara*, *Lactobacillus kunkeei*, *Bartonella apis*	*Serratia*, *Halomonadaceae*, *Gilliamella apicola*	32% of treated bees recovered 3 days after treatment	Tetracycline-treated bees have increased mortality

**Table 4 antibiotics-12-01438-t004:** Effect of RESERVE spectrum antibiotics on gastrointestinal dysbiosis.

Antibiotic	Experimental Model	Bacteria Decreased	Bacteria Increased	Time for Gut to Recover	Long Term Impacts
Omadacycline [73]	Human	*Bacteroides fragilis*, Bifidobacteria, Lactobacilli, and *Enterococcus* spp.	Lactose-fermenting *Enterobacteriaceae*	Within 3 weeks	Not Applicable

**Table 5 antibiotics-12-01438-t005:** ACCESS antibiotics’ mechanisms of dysbiosis.

Antibiotic	Mechanism of Dysbiosis	References
Doxycycline	Decrease in bacterial diversity	[57,59,62]
Tetracycline	Reduction of absolute bacterial abundance Increase in some opportunistic bacteria	[60,61]

**Table 6 antibiotics-12-01438-t006:** WATCH antibiotics’ mechanisms of dysbiosis.

Antibiotic	Mechanism of Dysbiosis	References
Oxytetracycline	Increase in opportunistic bacteria Decrease in microbial diversity and evenness	[63,64,65]
Minocycline	Decrease in bacterial diversity Failure to recover to pre-treatment levels in some bacteria Reduction in microbial richness Increase in opportunistic bacteria	[49,68,71,72]

**Table 7 antibiotics-12-01438-t007:** RESERVE antibiotics’ mechanisms of dysbiosis.

Antibiotic	Mechanism of Dysbiosis	References
Omadacycline	Decrease in total bacterial abundance Decrease in bacterial diversity Reduction of some species below limit of detection Increase in lactose-fermenting *Enterobacteriaceae*	[73]

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
