# Peer review of "Bidirectional Interaction between Tetracyclines and Gut Microbiome"

_antibiotics, 2023, doi:10.3390/antibiotics12091438_

Round 1
Reviewer 1 Report
Very well written review by Jaroszewski et al.
However it will be much better if the authors can include a Figure that summarizes the interaction between tetracycline and microbiome.
Minor editing is required.
Author Response
Journal: Antibiotics
Manuscript ID: antibiotics-2597810
Title of paper: Bidirectional interaction between tetracyclines and gut microbiome
Authors: Jerzy Jaroszewski, Niles Mamun, Krzysztof Czaja *
We would like to extend our heartfelt gratitude to you and the esteemed reviewers for their invaluable time and insightful comments on our manuscript. Being considered for publication in the prestigious Antibiotics Journal is an honor, and we genuinely appreciate the rigorous peer review process that has significantly enhanced the quality of our work. We have taken every reviewer's feedback to heart and revised our manuscript accordingly. In this response letter, we have addressed each reviewer's comments in a clear and organized point-by-point fashion. Furthermore, we have made all the requested modifications and incorporated additional information, as suggested by the reviewers. Below, we have provided detailed responses to every comment and suggestion, demonstrating our commitment to improving the manuscript. We look forward to the opportunity to contribute our research to the Antibiotics Journal, and we believe that the revisions we have made address the concerns raised during the peer review process effectively.
Thank you again for your time and consideration.
Sincerely yours,
Dr. Krzysztof Czaja
Response to comments from Reviewer # 1:
- Create a figure summarizing the interaction between tetracyclines and microbiome.
- Response: This figure has been added between lines 473 and 474.
Reviewer 2 Report
Overall, the manuscript is very well written and the topic is very interesting and was comprehensively reviewed
I have some suggestions /comments as follow
I) Introduction:
line 32
Please revise the wording “application” in the sentence “growing problem appears to most directly relate to excessive application”. I believe the authors meant the unnecessary intake but the word “application” may imply a topical route of administration and a valid indication.
II) General information about tetracyclines used in humans
Please consider briefly listing the different indications of tetracyclines. Probably best fit after the mechanism of action.
III) Tetracycline effects on the gut microbiome
Recommend including a table summarizing the mechanisms of tetracyclines induced dysbiosis
The paragraph from “375 to 380” is best suited at the end of the manuscript, maybe as future directions.
IV) How can the gut microbiome alter tetracycline treatment?
Recommend adding a figure (maybe GIT figure) to illustrate the different sites/ mechanisms where gut microbiomes compromise the efficacy of tetracycline or increase its toxicity. You can instead summarize in a table as suggested in section III
The paragraph about different forms of dysbiosis “430 to 435” doesn’t belong to this section but is rather a better fit in the beginning of section III.
The idea explained in lines “435 to 443” is better suited in the beginning of section IV before diving in expanding on these mechanisms.
In general, the manuscript might benefit from adding a section on how to mitigate the harmful effects of tetracyclines on gut microbiomes and vice versa including probiotics, food, etc.
But per minimum please add more discussion on how the combination with probiotics can mitigate the bidirectional detrimental influence of tetracycline on microbiomes. Please consider using the following publication and similar ones.
Montassier, E., Valdés-Mas, R., Batard, E., Zmora, N., Dori-Bachash, M., Suez, J. and Elinav, E., 2021. Probiotics impact the antibiotic resistance gene reservoir along the human GI tract in a person-specific and antibiotic-dependent manner. Nature microbiology, 6(8), pp.1043-1054.
The part about future research sounded repetitive in section II, III, and the conclusions.
Author Response
Journal: Antibiotics
Manuscript ID: antibiotics-2597810
Title of paper: Bidirectional interaction between tetracyclines and gut microbiome
Authors: Jerzy Jaroszewski, Niles Mamun, Krzysztof Czaja *
We would like to extend our heartfelt gratitude to you and the esteemed reviewers for their invaluable time and insightful comments on our manuscript. Being considered for publication in the prestigious Antibiotics Journal is an honor, and we genuinely appreciate the rigorous peer review process that has significantly enhanced the quality of our work. We have taken every reviewer's feedback to heart and revised our manuscript accordingly. In this response letter, we have addressed each reviewer's comments in a clear and organized point-by-point fashion. Furthermore, we have made all the requested modifications and incorporated additional information, as suggested by the reviewers. Below, we have provided detailed responses to every comment and suggestion, demonstrating our commitment to improving the manuscript. We look forward to the opportunity to contribute our research to the Antibiotics Journal, and we believe that the revisions we have made address the concerns raised during the peer review process effectively.
Thank you again for your time and consideration.
Sincerely yours,
Dr. Krzysztof Czaja
Response to comments from Reviewer # 2:
- I) Introduction:
- Line 32: Please revise the wording “application” in the sentence “growing problem appears to most directly relate to excessive application”. I believe the authors meant the unnecessary intake but the word “application” may imply a topical route of administration and a valid indication.
- Response: “Application” was changed to “prescription” to provide more clarity.
- II) General information about tetracyclines used in humans:
- Please consider briefly listing the different indications of tetracyclines. Probably best fit after the mechanism of action.
- Response: This paragraph was added from lines 183-201.
III) Tetracycline effects on the gut microbiome:
- Recommend including a table summarizing the mechanisms of tetracyclines induced dysbiosis.
- Response: This table was added between lines 384 and 393.
- The paragraph from “375 to 380” is best suited at the end of the manuscript, maybe as future directions.
- Response: This paragraph was moved to the end of the manuscript (494-499).
- IV) How can the gut microbiome alter tetracycline treatment?
- Recommend adding a figure (maybe GIT figure) to illustrate the different sites/ mechanisms where gut microbiomes compromise the efficacy of tetracycline or increase its toxicity. You can instead summarize in a table as suggested in section III.
- Response: This figure has been added between lines 473 and 475.
- The paragraph about different forms of dysbiosis “430 to 435” doesn’t belong to this section but is rather a better fit in the beginning of section III.
- Response: This section of text was moved to 215-220.
- The idea explained in lines “435 to 443” is better suited in the beginning of section IV before diving in expanding on these mechanisms.
- Response: Half this section of text was placed between 408 and 413, but the other half was moved between 444 and 448 as to not disrupt the flow of the paragraph.
- Please add more discussion on how the combination with probiotics can mitigate the bidirectional detrimental influence of tetracycline on microbiomes. Please consider using the following publication and similar ones. In general, the manuscript might benefit from adding a section on how to mitigate the harmful effects of tetracyclines on gut microbiomes and vice versa including probiotics, food, etc.
- Response: This paragraph was written at the end of section 4 at lines 460-472 and includes information from the suggested article.
Reviewer 3 Report

A review of the English language by a native speaker is recommended
Author Response
Journal: Antibiotics
Manuscript ID: antibiotics-2597810
Title of paper: Bidirectional interaction between tetracyclines and gut microbiome
Authors: Jerzy Jaroszewski, Niles Mamun, Krzysztof Czaja *
We would like to extend our heartfelt gratitude to you and the esteemed reviewers for their invaluable time and insightful comments on our manuscript. Being considered for publication in the prestigious Antibiotics Journal is an honor, and we genuinely appreciate the rigorous peer review process that has significantly enhanced the quality of our work. We have taken every reviewer's feedback to heart and revised our manuscript accordingly. In this response letter, we have addressed each reviewer's comments in a clear and organized point-by-point fashion. Furthermore, we have made all the requested modifications and incorporated additional information, as suggested by the reviewers. Below, we have provided detailed responses to every comment and suggestion, demonstrating our commitment to improving the manuscript. We look forward to the opportunity to contribute our research to the Antibiotics Journal, and we believe that the revisions we have made address the concerns raised during the peer review process effectively.
Thank you again for your time and consideration.
Sincerely yours,
Dr. Krzysztof Czaja
Response to comments from Reviewer # 3:
- The introduction is very dispersive. It is advisable to make it more fluid by highlighting the aim of the manuscript (the importance of investigating the bidirectional relationship between tetracyclines and the intestinal microbiome).
- Response: The aim of the manuscript was highlighted in the Introduction section.
- It is advisable to add a paragraph relating to literature studies conducted in order to develop strategies to minimize the negative effects of tetracyclines on the intestinal microbiome.
- Response: This paragraph was written at the end of section 4 at lines 460-472.
- It is advisable to check the references and always use the same format. For example, in reference 68 the DOI is missing.
- Response: Reference 68 has been corrected and all other references have been reviewed. Articles 47 and 55 do not have a DOI; a PMID was provided for them instead.
- A review of the English language by a native speaker is recommended.
- Response: English language was reviewed by a native speaker.
- The part about future research sounded repetitive in section II, III, and the conclusions.
- Response: Future research sections were reduced. The sentence that was at 444-446 has been removed. The sentence that was at 372-374 has been removed. The section of text that was at 468 to 472 has also been deleted and replaced.
Round 2
Reviewer 2 Report
All adequately addressed. I really liked the added figure.